**Data Availability Statement:** The whole genome sequencing data for this study (fastq files of all short read sequencing analyzes and five of the long read assemblies of the nine selected strains) have

# Predominance of multidrug-resistant *Salmonella* Typhi genotype 4.3.1 with low-level ciprofloxacin resistance in Zanzibar

Annette Onken[1,2,3]*, Sabrina Moyo[1,4], Mohammed Khamis Miraji[5], Jon Bohlin[6,7], Msafiri Marijani[8], Joel Manyahi[9], Kibwana Omar Kibwana[9], Fredrik Müller[10,11], Pål A. Jenum[3,11], Khamis Ali Abeid[12], Marianne Reimers[13], Nina Langeland[1,2], Kristine Mørch[1,2], Bjørn Blomberg[1,2]

**1** Department of Clinical Science, University of Medicine, Bergen, Norway, **2** National Centre for Tropical Infectious Diseases, Haukeland University Hospital, Bergen, Norway, **3** Department of Microbiology, Vestre Viken Hospital Trust, Drammen, Norway, **4** Department of Tropical Disease Biology, Liverpool School of Tropical Medicine, Liverpool, United Kingdom, **5** Department of Internal Medicine, Mnazi Mmoja Hospital, Zanzibar, Tanzania, **6** Department of methods and analysis, Section of modelling and bioinformatics, Domain of Infection Control, Oslo, Norway, **7** Center for Fertility and Health analysis, Norwegian Institute of Public Health, Oslo, Norway, **8** Pathology Laboratory Department, Mnazi Mmoja Hospital, Zanzibar, Tanzania, **9** Department of Microbiology and Immunology, Muhimbili University of Health and Allied Sciences, Dar es Salaam, Tanzania, **10** Department of Microbiology, Oslo University Hospital, Oslo, Norway, **11** Institute of Clinical Medicine, University of Oslo, Oslo, Norway, **12** Department of Pediatrics, Mnazi Mmoja Hospital, Zanzibar, Tanzania, **13** Emergency Care Clinic, Haukeland University Hospital, Bergen, Norway

* annetteonken@yahoo.com

## Abstract

### Background

Typhoid fever is a common cause of febrile illness in low- and middle-income countries. While multidrug-resistant (MDR) *Salmonella* Typhi (*S.* Typhi) has spread globally, fluoroquinolone resistance has mainly affected Asia.

### Methods

Consecutively, 1038 blood cultures were obtained from patients of all age groups with fever and/or suspicion of serious systemic infection admitted at Mnazi Mmoja Hospital, Zanzibar in 2015–2016. *S.* Typhi were analyzed with antimicrobial susceptibility testing and with short read (61 strains) and long read (9 strains) whole genome sequencing, including three *S.* Typhi strains isolated in a pilot study 2012–2013.

### Results

Sixty-three *S.* Typhi isolates (98%) were MDR carrying $bla_{TEM-1B}$, *sul1* and *sul2*, *dfrA7* and *catA1* genes. Low-level ciprofloxacin resistance was detected in 69% (43/62), with a single gyrase mutation *gyrA*-D87G in 41 strains, and a single *gyrA*-S83F mutation in the non-MDR strain. All isolates were susceptible to ceftriaxone and azithromycin. All MDR isolates belonged to genotype 4.3.1 lineage I (4.3.1.1), with the antimicrobial resistance

been deposited in the European Nucleotide Archive (ENA) at EMBL-EBI under accession number PRJEB59168 (https://www.ebi.ac.uk/ena/browser/view/PRJEB59168). The remaining four long read assemblies have been deposited in GenBank, National Institutes of Health (NIH), BioProject number PRJNA982791 (GenBank Overview (nih.gov)), accession numbers SAMN35714885 (ZNZ13L78), SAMN35713968 (ZNZ17F60), SAMN35714917 (ZNZ55M142), and SAMN35714939 (ZNZ57M158).). All other relevant data are within the manuscript, with the accession numbers of the genomes and additional information listed in the Supplementary information (S1 Table).

**Funding:** The study received funding from the Norwegian National Centre on Tropical Infectious Diseases, Haukeland University Hospital, Bergen, Norway. Vestre Viken Hospital Trust, Drammen, Norway supported the study including a salary to AO. Western Norway Regional Health Authority, University of Bergen, Norway supported AO by a PhD scholarship (Project Number 912277). CAMRIA - Combatting Anti-Microbial Resistance with Interdisciplinary Approaches, Centre for Antimicrobial Resistance in Western Norway, funded by Trond Mohn Foundation, supported the study (grant number TMS2020TMT11). SM has received a salary from STRESST - Antimicrobial Stewardship in Hospitals, Resistance Selection and Transfer in a One Health Context, University of Bergen, Norway (JPIAMR grant number NFR333432). The funders had no role in study design, data collection and analysis, decision to publish, or preparation of the manuscript.

**Competing interests:** The authors have declared that no competing interests exist.

determinants located on a composite transposon integrated into the chromosome. Phylogenetically, the MDR subgroup with ciprofloxacin resistance clusters together with two external isolates.

## Conclusions

We report a high rate of MDR and low-level ciprofloxacin resistant *S*. Typhi circulating in Zanzibar, belonging to genotype 4.3.1.1, which is widespread in Southeast Asia and African countries and associated with low-level ciprofloxacin resistance. Few therapeutic options are available for treatment of typhoid fever in the study setting. Surveillance of the prevalence, spread and antimicrobial susceptibility of *S*. Typhi can guide treatment and control efforts.

### Author summary

*Salmonella* Typhi causes typhoid fever. Multi-drug resistant (MDR) *S*. Typhi is spreading globally. Local and regional surveillance of MDR *S*. Typhi populations using both blood culture and whole genome sequencing can uncover outbreaks and help mapping the spread of *S*. Typhi and resistance mechanisms, which, in turn, can guide both control and prevention efforts and clinical management. Data regarding the distribution of MDR *S*. Typhi genotypes and resistance mechanisms is scarce in Zanzibar, Tanzania, as in many other African countries. In this study we characterize *S*. Typhi phenotypically and genotypically. This study shows a high rate of MDR *S*. Typhi, hence few therapeutic options are available for treatment of typhoid fever in the study setting. Our findings contribute to the knowledge base on typhoid fever in the region and to guide correct treatment of individual patients and control of the disease.

## Introduction

*Salmonella enterica* subspecies *enterica* serovar Typhi (*S*. Typhi) causes typhoid fever, an important global health problem with an estimated worldwide annual incidence of over 9 million cases, and about 100,000 to 200,000 deaths [1–4]. The estimated burden is uncertain due to diagnostic shortcomings and poor access to health care [5], but there are indications that the burden has declined lately [2]. In different geographic regions the incidence varies substantially, with incidence in excess of 800 cases per 100 000 persons per year in some settings in sub-Saharan Africa [6,7], and higher numbers in certain urban areas compared to rural settings [7]. Typhoid fever prevails in regions with poor sanitation facilities and limited access to clean drinking water [8]. Infections caused by multidrug-resistant (MDR) *S*. Typhi strains, defined by resistance to the prior first-line treatments ampicillin, chloramphenicol and trimethoprim-sulfamethoxazole, are complicated to treat and are associated with increased mortality [9,10]. The appearance of MDR *S*. Typhi in the 1970s led to widespread use of fluoroquinolones, and subsequent emergence of fluoroquinolone resistant *S*. Typhi in the early 1990s [11–13]. A combination of MDR and fluoroquinolone resistant *S*. Typhi leaves clinicians with few therapeutic options especially in developing countries. Since 2017, WHO ranked fluoroquinolone resistant *S*. Typhi as a high priority pathogen for research and development of new antibiotics [14].

Genetically *S*. Typhi is monomorphic [13]. A recent introduction of a phylogenetic genotyping scheme, GenoTyphi, has facilitated the interpretation of whole genome sequencing data (WGS) for *S*. Typhi [15–17]. Using this scheme, a global collection of *S*. Typhi isolates showed that the *S*. Typhi population is comprised of dozens of subclades which are specific for different geographical locations [15,17,18]. These global surveillance studies have shown that the majority of MDR *S*. Typhi infections worldwide belong to genotype 4.3.1 [18], which originated from South Asia and spread up to East Africa [15,17–19].

MDR in *S*. Typhi is linked to the presence of a composite transposon. In the 4.3.1. genotype, it was first introduced via the IncHI1-PST6 plasmid [18–21]. Later, reports have shown that the transposon carrying the genes associated with MDR was integrated into the *S*. Typhi chromosome [18,22,23]. This composite transposon carries antimicrobial resistance determinants which confer resistance towards penicillins (*bla*$_{TEM-1}$), trimethoprim (*dfrA7)* and sulfonamides (*sul1*, *sul2)*, chloramphenicol (*catA1)* and to streptomycin (*strA, strB)* [18,21,23].

Resistance against fluoroquinolones can result from mutations in the quinolone-resistance-determining regions (QRDRs) of the chromosomal *gyr* and *par* genes [10,24,25] and/or plasmid-mediated quinolone resistance (PMQR) which involves acquisition of e.g. *qnr* genes [24,26–28]. Using WGS, a large study [18] reported that the global population of MDR *S*. Typhi with reduced susceptibility to fluoroquinolones was associated with QRDR mutations. The MDR *S*. Typhi 4.3.1 subclade, which commonly also has reduced susceptibility to fluoroquinolones, is responsible for inter- and intra-continental spread, regional circulation, as well as local outbreaks in different parts of the world [18].

As the MDR *S*. Typhi population is increasing and spreading in different parts of the world, local/regional surveillance of MDR *S*. Typhi populations using both blood culture and WGS approach is important to report data on mechanisms responsible for resistance, for control and prevention of its spread, and to guide clinicians with available treatment options. However, in the African setting where typhoid fever is endemic, blood cultures for diagnostic confirmation and molecular characterization of *S*. Typhi are not performed routinely due to cost and infrastructure constraints [29]. Thus, there is paucity of data regarding the distribution of MDR *S*. Typhi genotypes and resistance mechanisms in Zanzibar, Tanzania, as in many other African countries. As part of a prospective study collecting blood-cultures from patients admitted with acute undifferentiated fever in Zanzibar, we here characterize *S*. Typhi phenotypically and genotypically to contribute to the knowledge base on typhoid fever in the region and to guide treatment and control.

## Methods

### Ethical statement

The research protocol was approved by the Zanzibar Medical Research and Ethical Committee (record no ZAMREC/0002/November/2014, renewal no ZAHREC/02/June/2019/41), and by the Regional Committee for Medical Research Ethics Health Region South East Norway (record no 2014/1940/REK South-East). Written informed consent was obtained from the patients or, in the case of children, from a parent or a responsible family member.

### Study site

Mnazi Mmoja hospital (MMH), Zanzibar, Tanzania, is the main referral hospital of the Zanzibar Archipelago with a population estimated to about 1.4 million in 2015 [30]. The 544-bed hospital also offers primary and secondary health care for the residents of Zanzibar city with a population of about 600,000 and is a teaching hospital for the State University of Zanzibar.

## Study population

Patients in the medical, pediatric and neonatal departments were enrolled in the study if they, either on admission or during their hospital stay, had fever ($\geq$38.3˚C in adults, $\geq$38.5˚C in children) or hypothermia (<36.0˚C), or were otherwise suspected to have systemic bacterial infection as judged by the clinician. Demographic and clinical information was obtained. A total of 1037 of 1043 eligible patients with fever and/or suspicion of serious systemic infection admitted to Mnazi Mmoja Hospital, Zanzibar, were consecutively included from March 17, 2015, to October 4, 2016 (in one patient two blood cultures were taken. Six patients were excluded because the blood culture sample was lacking). In addition, the three accessible (of in total seven) *S.* Typhi strains isolated during the pilot study in the years 2012 to 2013 [31] were included in the analyses. The details of the methodology of the pilot study have been previously described [31]. Briefly, the pilot study was performed at the same departments of Mnazi Mmoja Hospital as the main study, whereas 469 participants (neonates, children and adults) presenting with signs of systemic infections were included. As for the present study, the pilot study included clinical data, and blood was collected for culture and susceptibility testing of isolated microbes [31].

## Bacterial isolation and identification

Blood specimens were inoculated in BACTEC Myco/F lytic blood culturing vials (Becton Dickinson, Franklin Lakes, N.J.), one bottle per episode of febrile illness, incubated and analyzed as described earlier [31]. *Salmonella* Typhi isolates were identified by standard biochemical tests, the API 20E, VITEK 2 analysis using the identification cards for gram-negatives (both bioMérieux, Marcy-l'Etoile, France), and serogrouping by omnivalent A-67 and Vi-antigen (sifin diagnostics gmbh, Berlin, Germany). In the 61 isolates accessible for further analyses, the phenotypic identification was confirmed by whole genome sequencing.

## Antimicrobial susceptibility testing

Antimicrobial susceptibility testing for ampicillin, cefotaxime, ceftazidime, ciprofloxacin, trimethoprim-sulfamethoxazole, and chloramphenicol was performed by disc diffusion technique (Oxoid, Basingstoke, United Kingdom), and, for azithromycin, by minimal inhibitory concentration (MIC) gradient test (Liofilchem, Roseto degli Abruzzi, Italy) as described in the EUCAST guidelines [32]. Fifty-nine of the 61 isolates (the remaining two isolates were not frozen) were sent to Bærum Hospital, Vestre Viken Hospital Trust, Norway, for quality control of identification and susceptibility testing as well as for further characterization. For susceptibility testing, the same antimicrobials and the same disc diffusion technique as in Zanzibar were used. In addition, ceftriaxone, ciprofloxacin and azithromycin were tested by the MIC gradient test (Liofilchem). The three isolates of the pilot study were analyzed at the Department of Clinical Science, University of Bergen, Norway, using the same techniques. The results of the susceptibility testing were interpreted by the S-I-R system according to the EUCAST guidelines v 12.0 [33]. For the testing for ciprofloxacin, the EUCAST criteria for *Salmonella* species have been applied, classifying the strains with MIC >0.06 mg/L as resistant.

## Whole genome sequencing

Short read whole genome sequencing (WGS) was performed on 58 of the 61 *Salmonella* Typhi isolates (3 isolates not accessible) and on three *S.* Typhi strains of the pilot study [31]. For the 58 study strains, genomic DNA for the sequencing was extracted from single colonies using the Wizard genomic DNA Purification Kit (Promega, Madison, WI, USA) according to the

manufacturer's instructions. QIAxpert (QIAGEN, Valencia, CA, USA) was used for purity control. For ample library preparation TruSeq Nano reagents (Illumina, San Diego, CA, USA) were used. For the three strains of the pilot study, DNA isolation was performed using Invitrogen PureLink genomic DNA kit (Thermo Fisher Scientific, MA, USA) according to the manufacturer's recommendations. DNA library preparation was carried out using the standard protocol of the Nextera XT DNA Library Preparation Kit v. 3 (Illumina) up to the library amplification. Sequencing was performed on two runs of a MiSeq instrument (Illumina) with 300 bp paired end reads, according to the manufacturer´s instructions. All 61 short-read sequences were *de novo* assembled using the Pathogenwatch online platform (https://pathogen.watch). Fifty-eight of these 61 short-read sequences, all from the main study, were, in addition, assembled against a reference genome using Bowtie v. 2.3.4.2 [34], Samtools v.1.7 [35] and BCFTools v. 1.9 [35,36], using the largest closed genome representative for one of the two lineages as reference.

Long read sequencing was performed on six selected *S*. Typhi isolates of the main study (representing isolates with different antimicrobial resistance patterns and from both years 2015 and 2016), as well as three *S*. Typhi strains from the pilot study (31). DNA isolation was performed using PureLink genomic DNA kit (Thermo Fisher Scientific) according to the manufacturer's recommendations. Nanopore sequencing was performed using the rapid barcoding kit (SQK-RBK004) and MinION sequencer (Oxford Nanopore Technologies Ltd., Oxford, United Kingdom), and Guppy v. 3.2.10 for basecalling [37]. Hybrid-assembly was carried out using Unicycler v.0.4.8 [38] running with Pilon v. 1.23 [39] and Racon v. 1.4.3 [40] for error correction and sequence polishing on 'normal' settings. Five of the six isolates were completely assembled resulting in one contig, while isolate 50-M123 was returned with 14 contigs. Medusa v. 1.6 [41] together with one closed assembly from the different lineages was used to sort the 14 contigs so that they matched the other *Salmonella* isolates as closely as possible. Manual editing of contigs were subsequently carried out on the 50-M123 isolate to align sequence start with the other isolates in the study.

### Identification of resistance genes genotypes and sequence types

We used ResFinder (v4.1) with the default setting (90% threshold and 60% coverage) and MLST (v. 2.) with the default setting for the species, both from the Centre for Genomic Epidemiology CGE server (http://www.genomicepidemiology.org/), to identify acquired and chromosomal antimicrobial resistance determinants, and to assign sequence types, respectively. An online platform, Pathogenwatch (https://pathogen.watch) [42], was used to screen and assign *S*. Typhi genotypes. The Pathogenwatch tool uses GenoTyphi (code available at https://github.com/katholt/genotyphi) [42]. The GenoTyphi genotyping scheme divides the *Salmonella* Typhi population into four major lineages, and >75 different clades and subclades with the globally disseminated 4.3.1 (H58) subclade further delineated into lineages I and II (4.3.1.1 and 4.3.1.2) [15].

### Plasmid analysis

The presence of plasmids was investigated using PlasmidFinder v. 2.1 with the default setting (Danish Technical University, Denmark: http://cge.cbs.dtu.dk/services/PlasmidFinder/).

### Phylogenetic analysis

Conserved signature inserts phylogeny server (v. 1.4) (https://cge.food.dtu.dk/services/CSIPhylogeny/) was used to create a single nucleotide polymorphism (SNP)-based phylogenetic tree. All default values of the software were applied including minimum depth at SNP

positions 10x, minimum relative depth at SNP positions 10%, minimum SNP quality 30. The first phylogenetic tree compared the relatedness of our *S.* Typhi isolates, in the tree we included all 61 *S.* Typhi from this study which underwent WGS. The *S.* Typhi strain with accession number ERL12960 of genotype 4.3.1.1, isolated in India in 2012 [43] was used as a reference genome for the phylogenetic tree. The second phylogenetic tree was created to compare the 61 *S.* Typhi isolates from this study which belong to 4.3.1 genotype and map their relationship with other strains to get more insights into the spread of the 4.3.1 genotype. We included 38 published WGS sequences of *S.* Typhi of the same genotype from different parts of the world [18,22,28,44], including 16 isolates from Tanzania, fourteen from Kenya, two from Zambia, five from India and one from Pakistan, from the years 2007 to 2017. Both phylogenetic trees were annotated using the Interactive Tree of Life (v. 5.6.3) [45].

### Genetic environment for the MDR *S.* Typhi

The two representative *S.* Typhi from this study (ZNZ13L78 and ZNZ57M188) were annotated manually using a combination of RAST [46], Basic Local Alignment Search Tool (BLAST) (v. 2.11.0) [47], ResFinder (v. 4.1) [48] and MobileElementFinder (v. 1.0.3) from the CGE server and in SnapGene (v. 3.3.4) from GSL Biotech (available at snapgene.com). A comparative analysis using genoPlotr [49] was done for the chromosomal MDR gene segment of about 25kb of the two strains in this study and strain ERL12960.

## Results

During the study period of about 19 months in 2015–2016, 1038 blood cultures of 1037 patients were obtained, and 161 (16%) of these had growth of pathogens. *S.* Typhi was the most common pathogen, found in 61 of the 161 (38%) positive blood cultures (including nine patients with two and two patients with three isolates), corresponding to 35% (61/174) of the pathogens recovered. Additionally, among seven *S.* Typhi detected during the eight months pilot study in 2012–2013 [31], three isolates were available for further analyses. During the main study, the average isolation rate was more than three isolates per month compared to one isolate per month during the pilot study.

### Antimicrobial resistance pattern of *S.* Typhi

As shown in Table 1, MDR (defined as resistance towards ampicillin, trimethoprim-sulfamethoxazole and chloramphenicol) was observed in 98% of the isolates (63/64, including 60/61 from the main study and all three from the pilot study).

A total of 62 *S.* Typhi isolates (59 from the main study and three from the pilot study) were tested for ciprofloxacin resistance and 69% (43/62) showed low-level resistance with MIC values ranging from 0.125–0.25 mg/L. Of note, one of the *S.* Typhi isolates with low-level ciprofloxacin resistance was not MDR.

Results from disc diffusion showed all 64 *S.* Typhi isolates were susceptible to cefotaxime and ceftazidime. All tested isolates (fifty nine from the main study and three from the pilot study) were also susceptible to ceftriaxone by MIC gradient test with MIC values of 0.047–0.125 mg/L.

All 63 *S.* Typhi isolates were susceptible to azithromycin with MIC values 8–16 mg/L.

### Whole genome sequencing results

Short read sequencing was performed on 61 *S.* Typhi isolates (fifty-eight from the main study and three from the pilot study). A total of 60 isolates (98.3%) which phenotypically were MDR

**Table 1. Antimicrobial resistance pattern for *S.* Typhi isolates.**

| Antimicrobial agent | Number (%) of resistant isolates | | | | | | | |
|---|---|---|---|---|---|---|---|---|
| | Total | Genotype | | Year | | | | |
| | | 4.3.1.1 | 4.3.1.2 | 2012 | 2013 | 2015 | 2016 | |
| **Disc diffusion** | | | | | | | | |
| Ampicillin (n = 64) | 63* (98%) | 60 (100%) | 0 (0%) | 2 (100%) | 1 (100%) | 34 (100%) | 24 (96%) | |
| Chloramphenicol (n = 64) | 63* (98%) | 60 (100%) | 0 (0%) | 2 (100%) | 1 (100%) | 34 (100%) | 24 (96%) | |
| Trimethoprim-sulfamethoxazole (n = 64) | 63* (98%) | 60 (100%) | 0 (0%) | 2 (100%) | 1 (100%) | 34 (100%) | 24 (96%) | |
| Cefotaxime (n = 64) | 0* (0%) | 0 (0%) | 0 (0%) | 0 (0%) | 0 (0%) | 0 (0%) | 0 (0%) | |
| Ceftazidime (n = 64) | 0* (0%) | 0 (0%) | 0 (0%) | 0 (0%) | 0 (0%) | 0 (0%) | 0 (0%) | |
| **MIC gradient test** | | | | | | | | |
| Ceftriaxone (n = 62) | 0* (0%) | 0 (0%) | 0 (0%) | 0 (0%) | 0 (0%) | 0 (0%) | 0 (0%) | |
| Ciprofloxacin (n = 62) | 43* (69%) | 41 (68%) | 1 (100%) | 0 (0%) | 0 (0%) | 22 (65%) | 21 (84%) | |
| Azithromycin (n = 63) | 0* (0%) | 0 (0%) | 0 (0%) | 0 (0%) | 0 (0%) | 0 (0%) | 0 (0%) | |

* including 3 isolates of the pilot study 2012–2013

carried antimicrobial resistance determinants associated with resistance to ampicillin ($bla_{TEM-1}$), trimethoprim-sulfamethoxazole (*sul1* and *sul2* plus a *dfrA7* gene) and chloramphenicol (*catA1*). In addition to MDR determinants, these isolates also carried AMR determinants conferring resistance towards aminoglycosides (*aph(6)-Id* and *aph(3")-Ib*). One *S.* Typhi which was not MDR phenotypically, did not carry any AMR determinants. Table 2 shows the distribution of AMR determinants including QRDR mutations of the three different AMR genotype profiles. Detailed information on the 61 genomes is listed in S1 Table.

Resistance toward fluoroquinolones was associated with a single QRDR mutation in 42/43 *S.* Typhi which showed phenotypic low-level resistance to ciprofloxacin (the remaining strain was not accessible to WGS). Forty-one strains had single *gyrA*-D87G QRDR mutation, while one strain which was not MDR had a different single *gyrA*-S83F mutation.

As shown in Table 2, all MDR *S.* Typhi isolates were assigned to the 4.3.1.1 genotype using the GenoTyphi scheme [15,42]. On the contrary, one *S.* Typhi isolate (ZNZ50M123) which was not MDR was assigned to the 4.3.1.2 genotype. Furthermore, all *S.* Typhi isolates belong to MLST sequence type ST1.

## The genetic environment of resistance genes

Long-read sequencing was performed on six selected strains of the main study to investigate whether the AMR island is integrated in the chromosome or located on a plasmid. The gene contents of the five MDR *S.* Typhi isolates were highly similar, and the antimicrobial resistance determinants coding for MDR were located on a composite transposon integrated in the chromosome within a segment encompassed by insertion sequences and transposases. Fig 1 shows the structure of composite transposon of two *S.* Typhi isolates from this study ZNZ13L78 and ZNZ55M142 (with ENA accession numbers ERR11413875 and ERR11413524 respectively, for

**Table 2. Antimicrobial resistance genes and QRDR mutations of the three AMR genotype profiles.**

| AMR genotype profile | Aminoglycoside | Beta lactam | Phenicol | Sulfamethoxazole, trimethoprim | QRDR mutations |
|---|---|---|---|---|---|
| MDR/cipR sub-lineage of 4.3.1.1 | *aph(6)-Id, aph(3")-Ib* | $bla_{TEM-1B}$ | *catA1* | *sul1, sul2, dfrA7* | *gyrA*: p.D87G |
| MDR/cipS sub-lineage of 4.3.1.1 | *aph(6)-Id, aph(3")-Ib* | $bla_{TEM-1B}$ | *catA1* | *sul1, sul2, dfrA7* | - |
| nonMDR/cipR 4.3.1.2 isolate | - | - | - | - | *gyrA*: p.S83F |

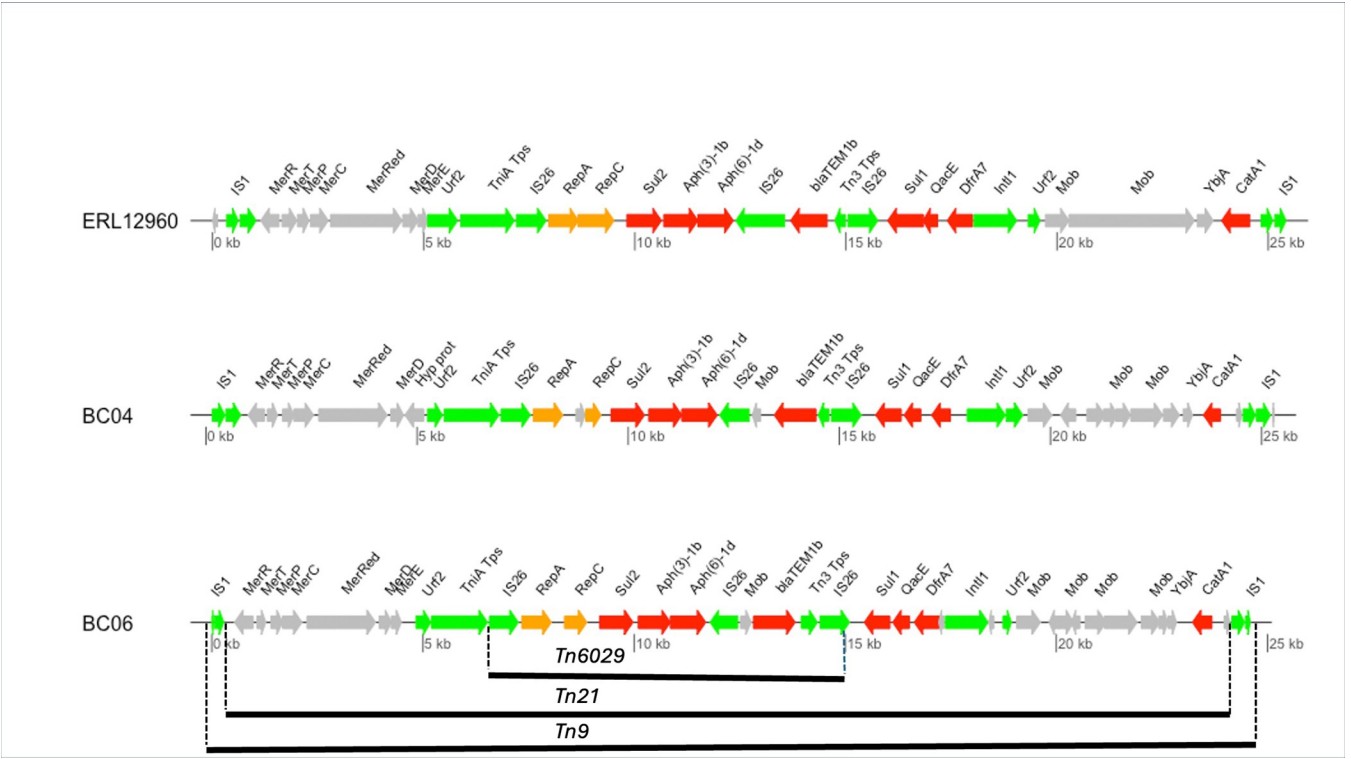

**Fig 1. The structure of MDR composite transposon.** The red arrows show antimicrobial resistance genes, MDR genes are *sul1/sul2*, *dfrA7*, *bla*$_{TEM-1}$ and *catA1*. Green arrows show insertion sequences (IS) or transposases and orange arrow show the replicons. BC04 is study isolate ZNZ13L78, BC06 is study isolate ZNZ55M142.

the short reads, and GenBank accession numbers SAMN35714885 and SAMN35714917 respectively, for the long reads) and ERL12960 as a comparison *S*. Typhi. All MDR *S*. Typhi isolates carried a composite transposon Tn*6029* (encoding *bla*$_{TEM-1B}$, *sul2*, *aph(6)-Id* and *aph (3")-Ib*) which was inserted in Tn*21* (carrying a class I integron encoding *dfrA* alleles in the gene cassette and *sul1*), which was in turn inserted within Tn*9* (encoding *catA1*) with IS*1* on both ends. No plasmid was found, but the MDR *S*. Typhi strains contained the IncQ1 plasmid replicon sequence (*repA* and *repC*) as shown in Fig 1. Furthermore, we detected chromosomal integration of the IS*1* at the known site downstream of *cyaY* as previously described [22].

## Phylogenetic analysis

Fig 2 is a whole genome SNP-based phylogenetic tree containing the 61 *S*. Typhi isolates from this study and one reference strain ERL12960. As shown in Fig 2, one isolate ZNZ50M123 (ENA accession numbers ERR11414360 and ERZ18316203, project *PRJEB59168*), highlighted in green belongs to genotype 4.3.1.2. This isolate has reduced susceptibility to ciprofloxacin (due to *gyrA*-S83F), and no acquired resistance genes. All other isolates were genotype 4.3.1.1, these were all MDR (due to *bla*$_{TEM-1}$, *catA1*, *sul1*, *sul2*, *dfrA7*). The majority of these belonged to a monophyletic clade (highlighted purple in tree) and had reduced susceptibility to ciprofloxacin (due to *gyrA*-D87G), herein referred as MDR/cipR. In contrast, the others (highlighted yellow in tree) including the older isolates from the pilot study, are fully sensitive to ciprofloxacin, and lack QRDR mutations referred as MDR/cipS. Within the study isolates, the SNP difference ranged between 0 and 41 (median 7.5), with the biggest SNP between the only isolate belonging to the 4.3.1.2 genotype (ZNZ50M123) and an isolate which is MDR/

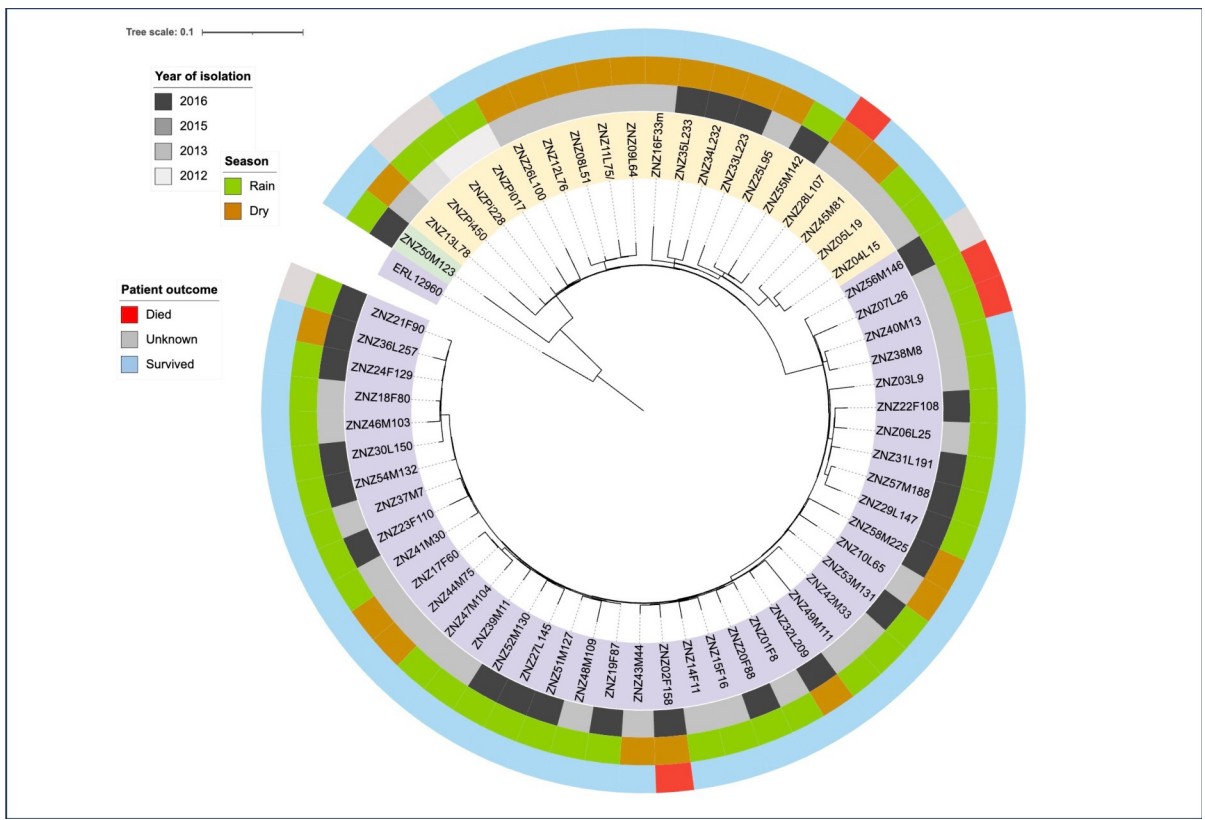

**Fig 2. Phylogenetic tree of *S*. Typhi isolates from this study using the reference genome ERL 12960 as midpoint root.** The inner circle shows three clusters of our *S*. Typhi: purple highlighted is a monophyletic clade (MDR/cipR), these are MDR and have reduced susceptibility to ciprofloxacin due to *gyrA*-D87G; yellow highlighted are strains which are MDR but sensitive to ciprofloxacin (MDR/cipS); one strain in green highlight is not MDR and has low-level ciprofloxacin resistance due to single *gyrA*-S83F QRDR mutation. The second circle shows the year of isolation of *S*. Typhi; black represents *S*. Typhi isolated in 2016, grey in 2015, light grey and very light grey were isolated in 2013 and 2012 respectively. The next circle shows the season when *S*. Typhi were isolated, green represents isolates collected during the rainy season while orange highlighted *S*. Typhi were collected during the dry season. The outer circle shows the outcome of the patients; red represents *S*. Typhi from patients who died, blue represents *S*. Typhi from patients who survived while grey represents *S*. Typhi from patients with unknown outcome.

cipR. For the genomes belonging to the 4.3.1.1 genotype, the overall difference of all study isolates ranged between 0 and 27 SNPs (median 7), within MDR/cipR isolates between 0 and 16 SNPs (median 6), within MDR/cipS isolates between 3 and 17 SNPs (median 9.5). The MDR/cipS isolates had a minimum difference of 11 SNPs to the closest neighbor of MDR/cipR isolates, with two of the pilot study isolates being closest related to MDR/cipR isolates. The only isolate of the genotype 4.3.1.2 (ZNZ50M123), had a difference between 25 and 41 SNPs (median 34) to the other study isolates (all belonging to 4.3.1.1 genotype).

In Fig 3, to provide context, the relationship of the 61 study strains to 38 strains from other publications including the reference strain ERL12960 is shown in a whole genome SNP-based phylogenetic tree. The minimum difference of the study isolates belonging to MDR/cipR group (genotype 4.3.1.1, MDR, and *gyrA*-D87G QRDR mutation) to isolates from other studies was 0 SNP in two isolates obtained in 2015 from travelers returning from Tanzania, both of which also were MDR and had identical *gyrA* point mutation, but no IncHI1 plasmid [28]. The 2nd most closely related isolate was also from Tanzania, and had 5 SNPs difference (year 2009, MDR, no IncHI1 plasmid, no FQ point mutation, ERR108659) [18]. The closest isolate

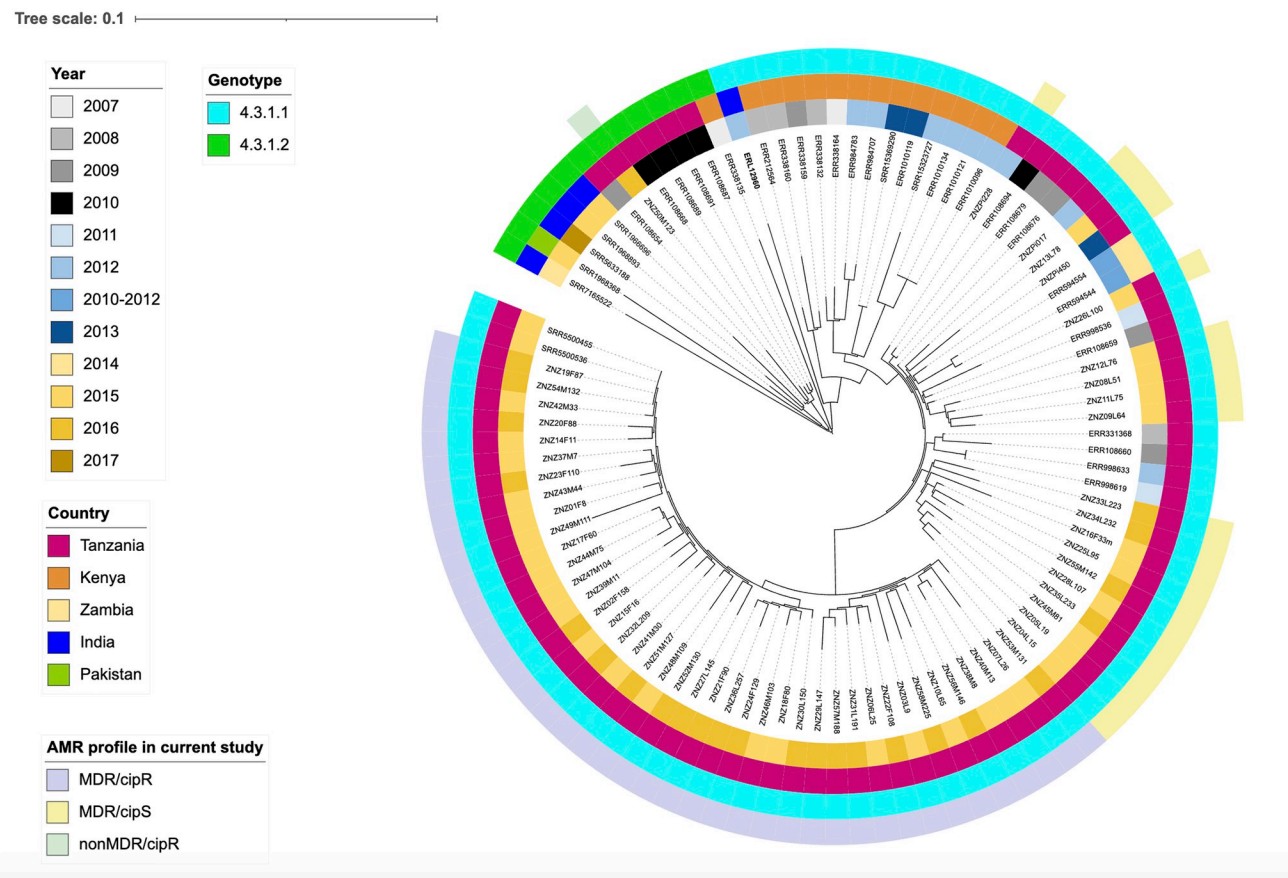

**Fig 3. Phylogenetic tree of *S*. Typhi isolates from this study and of isolates from other publications using the reference genome ERL 12960 as midpoint root.** From inside out, the inner circle shows the year of isolation of *S*. Typhi. The second circle shows the country where the strains were isolated. The next circle shows the genotype of the *S*. Typhi strains. The outer circle shows the AMR profile in the current study. The monophyletic MDR/cipR is highlighted in purple, the MDR/cipS cluster is yellow, and the only nonMDR/cipR strain is highlighted in green.

from Kenya had a difference of 17 SNPs. The isolates from Zambia had a difference of at least 15 SNPs, the minimum difference to isolates from Asia was 29 SNPs (India).

Isolates which are MDR/cipS had a minimum difference of 2 SNPs to isolates from East Africa (Tanzania, year 2009, with MDR, without the IncHI1 plasmid and without FQ point mutation) [18]. The closest isolate from Kenya had 11 SNPs difference. The difference to the Zambian isolates was at least 8 SNPs, and to the Asian isolates at least 22 SNPs (an isolate from India).

The non MDR genotype 4.3.1.2 isolate with the point mutation *gyrA*-S83F ZNZ50M123 had a minimum difference of 8 SNPs from another previously published Tanzanian isolate (year 2009, MDR, with IncHI1 plasmid and same *gyrA* point mutation, ERR108654) [18]. The second most closely related are four isolates from Tanzania with 10 SNPs distance (all from 2010, no MDR, no IncHI1 plasmid, all with the point mutation *gyrA*-S83F) [18]. The minimum distance to Kenyan isolates was 27 SNPs, to isolates from Zambia 29 SNPs, and to isolates from Asia (India) 21 SNPs.

All mentioned related external isolates belong to the ST1, as do the isolates from the current study.

## Discussion

The study shows a high rate of multidrug resistance in *S.* Typhi from Zanzibar throughout the study period, concurring with findings from previous studies from Tanzania [25], neighboring Malawi, Kenya and Zambia and other African countries [19,22,44,50]. The prevalence of typhoid fever increased from 1% (7/469) of patients in the pilot study in 2012/2013 to 6% (61/1037) in the current study from 2015/2016. Correspondingly, *S.* Typhi accounted for 9% (7/79) and 35% (61/174) of pathogenic bacteria in blood cultures in the pilot and the current study, respectively, suggesting an outbreak in 2015/2016. Our findings support the report of an ongoing epidemic of MDR *S.* Typhi in Africa [18]. The high prevalence of MDR and reduced susceptibility to fluoroquinolones limit the treatment options of typhoid fever in Zanzibar considerably.

In this study all MDR *S.* Typhi belong to genotype 4.3.1.1. This genotype has been linked with inter- and intra-continental spread of MDR *S.* Typhi [18]. Therefore, it is likely that genotype 4.3.1 has been imported from South Asia in the last two decades to East African countries including Tanzania, Kenya and Uganda [44] resulting in further spread of the genotype and local outbreaks. This genotype is associated with resistance to all three former first line drugs; ampicillin, chloramphenicol and trimethoprim-sulfamethoxazole. The availability over the counter and widespread use of these antibiotics in the study setting may exert a selective pressure that contributes to maintaining the persistence of MDR *S.* Typhi in the region. The only *S.* Typhi isolate which is not MDR belongs to genotype 4.3.1.2. This genotype has been reported in Southeast Asian countries such as India, Nepal, and Bangladesh [18], neighboring countries Kenya and Uganda [16,44,51], as well as in Tanzania [44].

Ciprofloxacin and other fluoroquinolones are widely used for the treatment of typhoid fever as a consequence of the high prevalence of MDR [52]. This study found that a high proportion of *S.* Typhi genotype 4.3.1.1 had low-level ciprofloxacin resistance. This is alarming as treatment with fluoroquinolones may lead to treatment failure [53] and relapse. Increased doses and prolonged therapy may be effective but may fuel emergence of high-level resistance [8]. In a multi-country typhoid fever surveillance study in Africa, isolates from Tanzania showed a high percentage of MDR (89%, 8/9 isolates), but no fluoroquinolone resistance was found [25]. However, the fluoroquinolone resistance rates currently increase on the African continent [24], and overuse of fluoroquinolones is likely partly to blame [54]. Fortunately, high-level fluoroquinolone resistance was not detected in the study strains, but continued widespread use of fluoroquinolone may bring it on.

In the present study, fluoroquinolone resistance was conferred by chromosomal single point mutations leading to structural alterations in topoisomerases as DNA gyrase. Increasing numbers of point mutations are correlated with a cumulative increase in MIC values, with the simultaneous acquisition of at least three-point mutations resulting in high-level fluoroquinolone resistance [24]. Changes in positions 83 and 87 of the *gyrA* gene are commonly reported point mutations [10,24]. All *S.* Typhi study isolates belonging to a monophyletic clade (MDR/cipR) had a single *gyrA*-D87G QRDR mutation and were phenotypically expressing low-level fluoroquinolone resistance. To our knowledge, in East Africa the *gyrA*-D87G point mutation has only been described in two *S.* Typhi strains isolated from returning travelers from Tanzania [28].

In contrast, the only isolate without MDR (isolate ZNZ50M123, belonging to the 4.3.1.2 genotype) harbors a different single mutation (*gyrA*-S83F), another common mutation in fluoroquinolone resistant *S.* Typhi, including East African isolates [25]. Furthermore, all *S.* Typhi isolates in this study were sensitive to azithromycin which would remain the only effective oral treatment for typhoid fever. Globally, azithromycin resistant *S.* Typhi is rare, but increasingly

reported in South-East Asia, notably in Bangladesh [55]. Increased use of azithromycin will subsequently pose a risk of introducing and spreading azithromycin-resistant *S.* Typhi in the African region [55]. Third generation cephalosporins such as ceftriaxone are still effective in Zanzibar, unlike in Pakistan, that currently experiences a long-lasting outbreak with an extensive drug-resistant *S.* Typhi strain, resistant to ceftriaxone, ciprofloxacin, ampicillin, trimethoprim-sulfamethoxazole, and chloramphenicol [52,55,56].

Earlier studies in Asia and some African countries have shown that MDR in *S.* Typhi is associated with the presence of an IncHI1 plasmid [18,44]. This means the main spread of antimicrobial resistance determinants has been horizontal transfer using a plasmid. Later studies have shown that MDR *S.* Typhi isolates from Zambia [22], Tanzania [44] and Asia [18,57] did not harbor plasmids associated with MDR determinants, suggesting that the genes conferring MDR have been incorporated in the chromosome of the bacteria. Concurring with previous findings [18,22,44], all MDR isolates in the present study carried a composite transposon integrated into the chromosome. Both the data from our study and from recent studies are compatible with spread of AMR through clonal expansion [18]. Previous studies suggest that the chromosomal location of the MDR determinants may confer a competitive advantage for the bacteria as it is less energy consuming compared to harboring a plasmid [28,44]. The integration of AMR determinants into the *S.* Typhi chromosome is worrying, as the chromosomal location reduces the likelihood of bacteria losing the antimicrobial resistance determinants [10]. The presumably low fitness cost associated with carriage of the MDR transposon [28] can provide a mechanism for sustained vertical transmission of MDR *S.* Typhi, even in the absence of selection pressure for the specific resistances [28].

In the phylogenetic comparison of the study isolates, monophyletic clade (MDR/cipR) isolates showed no SNP difference to the genotypes of *S.* Typhi isolated from two travelers returning to the United Kingdom from Tanzania during the same period [28]. They additionally share the same determinants coding for AMR including the identical *gyrA* point mutation, which, to our knowledge, has not been described in other *S.* Typhi isolates from East Africa. Isolates which are MDR/cipS, are closely related to other strains from Tanzania. Compared to MDR/cipR isolates, MDR/cipS has a smaller SNP difference to isolates from both Kenya and, especially, to those from Zambia. Assessing the results of the epidemiological and the antimicrobial resistance analyses, we speculate that the monophyletic clade (MDR/cipR) isolates may represent a new subtype and an outbreak strain, whereas MDR/cipS may be endemic.

The only non-MDR study strain is belonging to the genotype 4.3.1.2 and harboring a *gyrA*-S83F point mutation. It is closest related to a Tanzanian isolate from 2009 which had the same point mutation but differed by also harboring MDR and an IncHI1 plasmid. The second most closely related are four Tanzanian isolates from 2010 without MDR and without the IncHI1 plasmid, also with the same point mutation. In Asia, a decrease of MDR is associated with a corresponding decrease in carriage of IncHI1 plasmids [58], and we may speculate that the isolate from our study may also have lost the IncHI1 plasmid. This study isolate showed a smaller SNP difference to isolates from India than to isolates from neighboring Kenya or Zambia. This may support the earlier introduction and spread of a common ancestor from India to Tanzania.

In line with previous reports [42], the genetic findings in our study match the phenotypical results, emphasizing the potential utility of WGS for the prediction of AMR. The results are underlining that WGS is an important tool for surveillance of typhoid fever for uncovering outbreaks, and for understanding epidemiological relationships and the spread of antimicrobial resistance locally and globally. The high rate of MDR *S.* Typhi demonstrate the need of both antimicrobial stewardship for the treatment of suspected typhoid fever as well as surveillance. The study hospital is located on Unguja, the largest island of the Zanzibar Archipelago,

which has a multicultural population with historical links to mainland Tanzania, India and the Arabian Peninsula, in addition many international tourists visit the island. Extensive international travel may render the island vulnerable to the spread of resistant microbes [59], underlining the importance of continuous surveillance both locally and internationally.

## Conclusions

We report high rate of MDR and low-level ciprofloxacin resistant *S*. Typhi genotype 4.3.1.1 circulating in Zanzibar. The findings support that this clade now prevails in East Africa [16,44,50], leaving few therapeutic options available for treatment of typhoid fever in the setting. Surveillance of the prevalence, spread and antimicrobial susceptibility of *S*. Typhi can guide treatment and control efforts.

## Supporting information

**S1 Table. Per-isolate information on AMR genotype profile, year, AMR determinants and accession numbers of genomes (submitted to ENA project PRJEB59168 and to GenBank BioProject PRJNA982791).**
(DOCX)

## Acknowledgments

We thank all doctors, nurses, and all other staff at the Department of Internal Medicine and the Department of Pediatrics at Mnazi Mmoja Hospital, for their contribution to the study. We equally thank the technicians and all other staff of the Pathology Laboratory Department at Mnazi Mmoja Hospital who facilitated and contributed to the study. We are also grateful the support by Merriam Sundberg-Amargo and Harald Landa for their help in preparing the shipment of material from Norway to Zanzibar as well as by Inger Marie Brend Johansen, Mette Voldhaug and Tone Kofstad for the further analysis of the strains, all from the Department of Microbiology at Vestre Viken Hospital Trust, Norway.

The Illumina sequencing service of the study strains was provided by the Norwegian Sequencing Centre (www.sequencing.uio.no), a national technology platform hosted by Oslo University Hospital and the University of Oslo, supported by the "Research Council of Norway and the Southeastern Regional Health Authorities" and, for the three isolates of the pilot study, by Akershus University Hospital, Lørenskog, Norway.

The Illumina short read sequencing of three strains of the pilot study as well as the MinION sequencing (six strains of the main study and three strains of the pilot study) was provided by Akershus University Hospital, Lørenskog, Norway.

Jessin J. J. Peter, University of Tromsø, Norway, performed a first analysis of the WGS data.

Thank you to R. S. Hendriksen, Technical University of Denmark, regarding the advice for the analysis by long read sequencing (MinION).

## Author Contributions

**Conceptualization:** Annette Onken, Nina Langeland, Bjørn Blomberg.

**Data curation:** Annette Onken, Sabrina Moyo, Jon Bohlin, Bjørn Blomberg.

**Formal analysis:** Annette Onken, Sabrina Moyo, Jon Bohlin, Bjørn Blomberg.

**Funding acquisition:** Annette Onken, Nina Langeland, Kristine Mørch, Bjørn Blomberg.

**Investigation:** Annette Onken, Sabrina Moyo, Mohammed Khamis Miraji, Jon Bohlin, Msafiri Marijani, Kibwana Omar Kibwana, Khamis Ali Abeid, Marianne Reimers, Bjørn Blomberg.

**Methodology:** Annette Onken, Sabrina Moyo, Jon Bohlin, Fredrik Müller, Pål A. Jenum, Bjørn Blomberg.

**Project administration:** Annette Onken, Msafiri Marijani, Bjørn Blomberg.

**Resources:** Annette Onken, Sabrina Moyo, Mohammed Khamis Miraji, Jon Bohlin, Msafiri Marijani, Kibwana Omar Kibwana, Pål A. Jenum, Khamis Ali Abeid, Kristine Mørch, Bjørn Blomberg.

**Software:** Annette Onken, Sabrina Moyo, Jon Bohlin, Bjørn Blomberg.

**Supervision:** Annette Onken, Mohammed Khamis Miraji, Kibwana Omar Kibwana.

**Validation:** Annette Onken, Sabrina Moyo, Jon Bohlin, Nina Langeland, Bjørn Blomberg.

**Visualization:** Annette Onken, Sabrina Moyo, Bjørn Blomberg.

**Writing – original draft:** Annette Onken.

**Writing – review & editing:** Annette Onken, Sabrina Moyo, Mohammed Khamis Miraji, Jon Bohlin, Msafiri Marijani, Joel Manyahi, Kibwana Omar Kibwana, Fredrik Müller, Pål A. Jenum, Khamis Ali Abeid, Marianne Reimers, Nina Langeland, Kristine Mørch, Bjørn Blomberg.

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
