## [Decision Letter · Decision Letter 0]

8 Mar 2024

Dear Ms. Onken,

Thank you very much for submitting your manuscript "Predominance of multidrug-resistant *Salmonella* Typhi genotype 4.3.1 with low-level ciprofloxacin resistance in Zanzibar" for consideration at PLOS Neglected Tropical Diseases. As with all papers reviewed by the journal, your manuscript was reviewed by members of the editorial board and by several independent reviewers. The reviewers appreciated the attention to an important topic. Based on the reviews, we are likely to accept this manuscript for publication, providing that you modify the manuscript according to the review recommendations. 

Sincerely,

Joseph M. Vinetz

Section Editor

Joseph Vinetz

Section Editor

Reviewer's Responses to Questions

**Key Review Criteria Required for Acceptance?**

**Methods**

-Are the objectives of the study clearly articulated with a clear testable hypothesis stated?

-Is the study design appropriate to address the stated objectives?

-Is the population clearly described and appropriate for the hypothesis being tested?

-Is the sample size sufficient to ensure adequate power to address the hypothesis being tested?

-Were correct statistical analysis used to support conclusions?

-Are there concerns about ethical or regulatory requirements being met?

Reviewer #1: this article focuses on the genetic makeup of MDR S.typhi. the topic is relevant and of much importance at global level. The objectives are clear and well-articulated. the study design is fine but lacks details about the pilot project from where historical samples were assessed and compared. the isolates included in this analysis are from 2016, would have been nice to analyze the currently prevailing isolates, its almost 8 years old data. Nevertheless, the information adds value to the scientific literature. Appropriate Ethical compliance is in place.

Reviewer #2: The objectives are clear and the study design and methodology are appropriate and well described.

**Results**

-Does the analysis presented match the analysis plan?

-Are the results clearly and completely presented?

-Are the figures (Tables, Images) of sufficient quality for clarity?

Reviewer #1: the analysis is quite complete and well presented. Tables and Fig are well presented with detailed legends appropriate for the scientific audience.

Reviewer #2: 1. The results match the aims and methods, however I found the description of the key findings on population structure very hard to follow. In particular the discussion of the various groups very hard to follow, because there is no one table/figure/paragraph that links the ‘group’ labels; genotypes; and AMR. Also because the tree topologies are not rooted correctly, it is hard to see but I think the cluster described as ‘group A’ is actually a monophyletic lineage emerging from within ‘group B’. 

Reading closely and looking at the figures, I think the population structure could be defined much more simply and clearly as follows:

- One isolate (ZNZ50M123) is genotype 4.3.1.2, this has reduced susceptibility to ciprofloxacin (due to gyrA-S83F), and no acquired resistance genes.

- All other isolates were genotype 4.3.1.1, these were all MDR (due to blaTEM-1, catA1, sul1, sul2, dfrA7)

- The majority of these belonged to a monophyletic clade (purple in tree) that had reduced susceptibility to ciprofloxacin (due to gyrA-D87G).

- The others (yellow in tree) include the older isolates from the pilot study, are fully sensitive to ciprofloxacin, and lack QRDR mutations.

Adding the genotypes and years to Table 1 would further clarify; as would rooting the trees properly to highlight the nature of the MDR/CipR sublineage better.

In my view, this provides greater clarity for readers in fewer words, and without needing to introduce arbitrary subgroup labels (group C is a single isolate, so does not need a name; group A can be described as “the local MDR/cipR sub lineage of 4.3.1.1”, which you could give a label to if you really want to; group B is just the rest, it is not monophyletic and doesn’t really need a name (if wanting to refer to these if could be described simply as “MDR/cipS 4.3.1.1”.

I would strongly encourage the authors to rework the manuscript to consider these points.

2. Table 2 lists all AMR determinants in all 65 genomes (ie one row per isolate), this large table is useful but could be moved to Supplementary and replaced with a summary table in the main text, with one row per AMR genotype profile. The per-isolate table would benefit from adding genotype, accession, year, etc

3. The results concerning integration of MDR are well described except that the authors don’t say what the integration site is - ie where in the chromosome is it integrated. Is it one of the previously described sites, or a novel one?

4. Lines 334-336, “No plasmid was found, but the MDR S. Typhi strains contained the IncQ1 plasmid replicon sequence (repA and repC) as shown in Figure 1. These are probable remnants of IncHI1 integrated in the chromosome.” - the repA/repC is a part of the composite transposon it is not derived from IncHI1

5. How are Figures 2A and 2B rooted? The topologies are very hard to see with the current rooting and plotting choices. Standard practice is to use an outgroup root, or midpoint root; either would solve the issue in this case.

6. Figure 2B - the colour palettes here are very hard to follow. Suggest changing the years to a greyscale (lighter = earlier, darker = later) as was done in Figure 2A.

**Conclusions**

-Are the conclusions supported by the data presented?

-Are the limitations of analysis clearly described?

-Do the authors discuss how these data can be helpful to advance our understanding of the topic under study?

-Is public health relevance addressed?

Reviewer #1: yes, the conclusions are supported by the data presented in the result section.

Reviewer #2: The conclusions are well summarised and supported by the data.

**Editorial and Data Presentation Modifications?**

Reviewer #1: some details regarding the pilot project must be mentioned for readers to know the relevance. need to mention why recent isolates were not included in the analysis. data comprises of isolates from 2015-2016, that is almost 8 years old data. data linked to recent isolates would make it more valuable.

Reviewer #2: (No Response)

**Summary and General Comments**

Reviewer #1: good work relevant to the local community and globally as the MDR and XDR S.typhi continue to rise, molecular epidemiology is important in tapping the movements of these bugs and assessing the regional genetic mutation pressure. weakness: data is 8 years old, addition of current isolates to the analysis will add value to the data and publication.

Reviewer #2: The article is overall well written and presents novel data on the Typhi pathogen population responsible for typhoid fever in Zanzibar. The methods for characterisation (AST, WGS) are state of the art and fully described. Importantly, the authors place their findings regarding the Zanzibar population in the global context, by comparison with public genomic data.

I noted two issues that should be addressed in the Introduction:

1. Lines 99-101 refer to prior work describing the global population structure of Typhi, however the papers referenced are nearly a decade old (2015/16), suggest citing the 2023 update from the Global Typhoid Genomics Consortium which describes 13,000 genomes (Carey et al 2023, eLife https://doi.org/10.7554/eLife.85867)

2. Line 105, “MDR in S. Typhi is linked to the presence of an integrative conjugative element (ICE).” This statement is not supported by a reference and is incorrect, to my knowledge. MDR in Typhi is associated with a transposon, mobilised by IS1 via a non-conjugative mechanism. The transposon often resides on a conjugative plasmid, but is not itself a conjugative element.

PLOS authors have the option to publish the peer review history of their article (what does this mean?). If published, this will include your full peer review and any attached files.

Reviewer #1: No

Reviewer #2: No

Figure Files:

Data Requirements:

Reproducibility:

References

---

## [Editor Report · Decision Letter 1]

2 Apr 2024

Dear Ms. Onken,

We are pleased to inform you that your manuscript 'Predominance of multidrug-resistant *Salmonella* Typhi genotype 4.3.1 with low-level ciprofloxacin resistance in Zanzibar' has been provisionally accepted for publication in PLOS Neglected Tropical Diseases.

Best regards,

Joseph M. Vinetz

Section Editor

Joseph Vinetz

Section Editor

---

## [Editor Report · Acceptance letter]

11 Apr 2024

Dear Ms. Onken,

We are delighted to inform you that your manuscript, "Predominance of multidrug-resistant *Salmonella* Typhi genotype 4.3.1 with low-level ciprofloxacin resistance in Zanzibar," has been formally accepted for publication in PLOS Neglected Tropical Diseases.

Best regards,

Shaden Kamhawi

co-Editor-in-Chief

Paul Brindley

co-Editor-in-Chief
